# Advances in the Study of Olfaction in Eusocial Ants

**DOI:** 10.3390/insects12030252

**Published:** 2021-03-17

**Authors:** Stephen T. Ferguson, Isaac Bakis, Laurence J. Zwiebel

**Affiliations:** Department of Biological Sciences, Vanderbilt University, Nashville, TN 37235, USA; stephen.t.ferguson@vanderbilt.edu (S.T.F.); isaac.bakis@vanderbilt.edu (I.B.)

**Keywords:** Hymenoptera, Formicidae, ants, eusocial, olfaction, odor coding, review

## Abstract

**Simple Summary:**

In contrast to solitary insects such as the fruit fly or mosquito, eusocial ants form colonial societies comprised of reproductives (female queens and short-lived males) and thousands of sterile female offspring known as workers. Social behaviors such as nursing the queen’s offspring, foraging for food, and nest defense emerge from the collective behavior of the workers. Importantly, these behaviors critically depend on the exchange of information through the detection of chemical cues in the environment. While research efforts have historically aimed to understand the chemical ecology of these social insects, advances in the application of molecular technology over the past decade have facilitated novel studies that bridge the gap between social behaviors and olfaction. Here, we review major advances in the study of molecular olfaction in eusocial ants and highlight potential avenues for future research.

**Abstract:**

Over the past decade, spurred in part by the sequencing of the first ant genomes, there have been major advances in the field of olfactory myrmecology. With the discovery of a significant expansion of the odorant receptor gene family, considerable efforts have been directed toward understanding the olfactory basis of complex social behaviors in ant colonies. Here, we review recent pivotal studies that have begun to reveal insights into the development of the olfactory system as well as how olfactory stimuli are peripherally and centrally encoded. Despite significant biological and technical impediments, substantial progress has been achieved in the application of gene editing and other molecular techniques that notably distinguish the complex olfactory system of ants from other well-studied insect model systems, such as the fruit fly. In doing so, we hope to draw attention not only to these studies but also to critical knowledge gaps that will serve as a compass for future research endeavors.

## 1. Introduction

### 1.1. A Remarkable Olfactory Sense

From an evolutionary perspective, ants are an extraordinarily successful insect taxon that are globally pervasive and comprise more than a quadrillion individuals [1]. This success is likely due to the complex eusocial structure and sophisticated olfactory system that drives collective behavior among individuals throughout a colony (Figure 1). Without centralized control, sterile female workers tend to the queen’s offspring (“nurses”), construct and maintain nests (“builders/midden workers”), defend and police the colony (“soldiers”), and search for food (“foragers”). Beyond these fundamental tasks, which are commonly observed across eusocial insect taxa, the social life of certain ant species may be quite extraordinary. Attine ants rely on the collection of leaves that they use as a substrate to maintain elaborate fungal gardens [2]. Army ants create living nests with their bodies, known as bivouacs, where they shelter the queen and store food and brood among the interior chambers [3]. When selecting a new nest site, rock ants engage in a democratic decision-making process that relies on quorum sensing [4]. To accomplish these impressive feats, ants rely largely on sophisticated chemical communication systems that provide an extraordinary degree of discrimination and sensitivity [5].

Ants communicate with one another by exchanging an array of chemical messages that includes general odorants common to many insects, as well as the social pheromones and other chemical blends that distinguish ants from other, solitary insects. Many of these messages are detected via olfactory signal transduction pathways largely localized to the antennae [5]. Complex blends of cuticular hydrocarbons (CHCs), as an example, are an especially important class of semiochemicals that convey a broad range of social information including colony membership, fertility, and task group [6]. In the course of a brief antennation event, where ants make mutual contact with their antennae, an ant can identify a foraging nestmate or an intruding non-nestmate based on their respective CHC profiles [7,8]. CHCs are produced by oenocytes associated with the fat body [9,10,11], and it is believed that the post-pharyngeal gland plays a central role in storing and distributing the hydrocarbons involved in colony identity [12]. Indeed, there are considerable qualitative and quantitative similarities between the contents of the post-pharyngeal gland and CHC profiles [13]. Taken together, these studies highlight only a small fraction of the complex pheromone biochemistry responsible for the organization and coordination of ant societies.

Over and above the role of CHCs, ants have been described as “walking chemical factories” because they rely on a large array of exocrine glands that collectively produce the semiochemical releasers for many complex social behaviors [1]. For example, in *Formica argentea*, undecane is produced in high concentrations in the Dufour’s gland, where it is likely to act as an alarm pheromone component [14]. In addition to its role in predation and defense, the poison gland of Formicidae produces formic acid, which may act synergistically with other compounds that elicit alarm responses in the Dufour’s gland [15,16]. Even more notorious is fire ant venom, which is comprised of hydrophobic dialkylpiperidines, known as solenopsins, used for predation and defense [17]. Moreover, even closely-related species may have strikingly different exocrine gland composition. This is illustrated in studies that examined the phylogenetic relationship of *Camponotus floridanus* to *C. atriceps*, one which was contested for a time with some suggesting that the two species were synonymous [18]. However, the ratio of compounds in the Dufour’s gland was observed to be notably different, with certain compounds, such as 2-methyldecane and heneicosane, present in only one species or the other [19]. This distinct phylogeny is also consistent with studies demonstrating that trail-following behaviors are evoked by distinct hindgut components found in each species. In this regard, *C. floridanus* is sensitive to nerolic acid, while *C. atriceps* relies on 3,5-dimethyl-6-(1’-methylpropyl)-tetrahydropyran-2-one [19]. In short, there is tremendous diversity of exocrine gland form and function among ants, including glands that may elicit behaviors that are unique to a given genera [1].

### 1.2. Aim and Scope of This Review

The availability of the first ant genome sequences [20] revealed that ants have a greatly expanded odorant receptor (OR) gene family compared with other, solitary insects [21,22]. Such studies are helping to bridge the gap between ant chemical ecology and the underlying molecular machinery responsible for these complex eusocial interactions. While there is a considerable body of literature on the complex lifecycles and biology of ants, this review will focus on what we consider to be the major advances in the study of ant olfactory systems and their role in mediating that biology. In doing so, our intent is to go beyond a simple accounting of these efforts and to highlight several avenues for future studies that will address critical knowledge gaps to provide a better understanding of the fundamental aspects of eusocial insect biology.

## 2. The Peripheral Olfactory System

The complex array of sensory neurons and support cells that together make up the peripheral olfactory system is the initial site of chemical detection and perhaps discrimination in ants. Here, pheromones, kairomones, and other semiochemicals are detected by an array of membrane-bound chemoreceptors expressed in suites of olfactory sensory neurons (OSNs). The function of these OSNs relies on a spectrum of signal transduction pathways that comprise both extra- and intracellular components centered around three classes of transmembrane chemoreceptors: ORs, gustatory receptors (GRs), and ionotropic receptors (IRs) (for a detailed discussion of these and other chemoreceptor proteins involved in Hymenopteran olfactory biology, we direct the reader to [5]). In brief, individual ORs are expressed in a subset of OSNs alongside the obligate and highly conserved odorant receptor co-receptor (Orco) [23]. The ORs are involved in the detection of pheromones and other general odorants which for ants notably include the CHCs [24,25]. ORs are hypothesized to derive from GRs, which are the most ancient chemoreceptor family in insects and, at least in *Drosophila*, are responsible for the direct (contact-based) detection of tastants as well as carbon dioxide [26]. Curiously, while empirical evidence suggests that ants are able to detect carbon dioxide [27,28,29], they have lost the canonical CO_2_ receptors found in dipteran species [21,30]. The IRs are derived from ancestral glutamate receptors and form an independent lineage of chemoreceptors that are, in *Drosophila*, responsible for the detection of acids and aldehydes [31,32,33]. Beyond these primary chemoreceptor families, there are a number of other ancillary support proteins involved in olfactory signaling. These include odorant degradation enzymes (ODEs) and a variety of odorant binding proteins (OBPs) and chemosensory proteins (CSPs). The former, as their name implies, facilitate the degradation of odorants [34,35], while the function of the latter is not as well understood. However, it is commonly thought that the OBPs and CSPs may facilitate odor transport through the sensilla lymph [36], although they are evidently not required for olfactory responsiveness [37,38].

The OSNs subtend hair-like sensilla that are stereotypically distributed along the antennae and other chemosensory appendages [39]. For ants, there are several different types of sensilla that vary in function, innervation, and morphology [40,41]. These notably include basiconic (broadly chemoreceptive including CHCs), ampullaceal (putatively CO_2_ receptive), chaetic (contact-based chemosensation), coelocapitular (hygro- and thermoreceptive), coeloconic (chemoreceptive), and trichoid (chemoreceptive) sensilla. In contrast to the relatively simple Dipteran olfactory system, which may have only a handful of OSNs in each sensillum, ant sensilla may contain over 130 OSNs [40]. In addition, there are important sexual dimorphisms with respect to the broad morphology of the antennae and the composition of sensilla between female and male ants. The basiconic sensilla, which presumably house the OSNs involved in CHC detection [42,43], are notably absent in males and likely reflect the distinct physiological function of behavior of the different members of the colony [40,44,45]. Altogether, the peripheral olfactory system in ants shares many features in common with other insect species; however, evolution has produced an unparalleled level of complexity in ants that is unrivaled even by their much more studied Dipteran counterparts (Figure 2).

### 2.1. Untangling Odor Coding in the Peripheral Sensilla

While there is a rich body of literature describing the source and function of pheromones and other semiochemicals that regulate the collective social behaviors in ant colonies, considerably less is known about odor coding in the antennae and other olfactory appendages. One area of particular interest is the characterization of the olfactory processes involved in translating the CHC signatures that underlie nestmate recognition, whereby conspecific ants are able to discriminate workers from their home colony (nestmates) which are met passively as “friends” from workers from other colonies (non-nestmates) which are usually treated aggressively as “foes”. This has proven to be exceptionally challenging to address due to the complex CHC blends that are utilized along with the combinatorial and multifaceted nature of CHC detection. At first glance, the CHC profiles of conspecific workers from different colonies are often qualitatively similar, differing only in the subtle quantitative differences in the proportion of a given hydrocarbon [43,46,47,48]. Despite these seemingly imperceptible differences, many species of ants are robustly able to use this information to distinguish friends from foes [49], identify the task of a fellow nestmate [50], and discriminate aberrant worker-laid eggs from those of their queen [51]. This remarkable sensory acuity is accomplished, at least in part, through CHC detection by the multiporous basiconic sensilla [43]. The odor coding in the ant basiconic sensilla remains enigmatic due to the astonishingly high number of OSNs (>130) that are present [40] and may be interacting with each other either directly via gap junctions [52] or indirectly via ephaptic transmission [53]. Furthermore, as of this writing, the precise composition of chemoreceptors expressed by these diverse OSNs remains unknown, although there are at least three subtypes of basiconic sensilla in two *Camponotus* species that collectively detect more than 10 general odorants and at least 20 different hydrocarbons [43]. Taken together, the sheer diversity of stimuli, as well as the range of interacting neuronal and molecular receptors, represent a profoundly complex odor-coding process that is likely to be beyond our understanding for quite some time to come.

While challenging, deciphering at least some of the linkage between the subtle complexities with which information is encoded in CHC profiles and the densely packed OSNs in the basiconic sensilla will undoubtedly represent a substantial milestone in olfactory myrmecology. To that end, several conflicting hypotheses have been proposed and experimentally examined. Single sensillum recordings (SSRs) showing that *C. japonicas* workers only respond to non-nestmate (but notably not nestmate) CHC blends led to the suggestion that ants may be anosmic to their own colony odor [42]. If this effect were broadly observed, this would be remarkable because the hydrocarbons comprising nestmate and non-nestmate CHC blends are presumably the same, differing only in their ratio [43,46,47,48]. However, subsequent studies using both SSR and antennal lobe (AL) activity imaging have not replicated these findings as both nestmate and non-nestmate CHCs were detected in the antennae and AL glomeruli, respectively [43,54,55]. A number of attempts have been made to reconcile these discordant findings. For example, it has been suggested, but as yet not validated experimentally, that there are at least two sensilla subtypes: one dedicated to detecting non-nestmate CHCs and another that detects a broad spectrum of hydrocarbons [56]. As things stand, we are left with more questions than answers, such that peripheral and central odor coding in eusocial insects remain largely hypothetical.

### 2.2. Identifying Odor Ligands through the Deorphanization of Chemoreceptors

Compared with the vast literature on ant pheromone biochemistry and chemical ecology, far less is known about the chemoreceptors involved in the detection of these odorants. Several efforts to functionally characterize (a process sometimes referred to as “deorphanization”) the diverse spectrum of ant ORs which comprise over 20 phylogenetically related subfamilies [21,41] through the identification of their biologically salient odor ligands have been carried out. While initial deorphanization studies identified the receptor for 4-methoxyphenylacetone in *H. saltator* (HsOr55) and 2,4,5-trimethylthiazole in *C. floridanus* (CfOr263) [21], the two most notable studies in this regard were conducted in *H. saltator* [24,25] where the electrophysiological responses of 47 ORs across nine different subclades were examined against a panel of synthetic and naturally obtained hydrocarbons and a range of other general odorants. These studies revealed that, while the rapidly-evolving 9-exon OR subfamily is able to detect CHCs and, therefore, remains a compelling aspect of Hymenopteran olfaction, it is clear that pheromone detection is not limited to this subfamily. While several members of the large 9-exon subfamily—HsOr263, HsOr271, and HsOr259-L2—were indeed responsible for the detection of 13,23-dimethylheptatriacontane, a putative queen pheromone, responses to other hydrocarbons, as well as a range of general odorants, were broadly detected across the various subclades. These notably included an OR (HsOr36; enriched in males) from the L subfamily, a subfamily H receptor (Hs210; enriched in workers), a subfamily V receptor (HsOr170), and a subfamily E receptor (HsOr236), all of which respond to long-chain alkanes.

These studies are important because they contribute to our understanding of the evolution of olfactory function in social insects [57], yet these relatively modest efforts only scratch the surface. Future studies on the functional characterization of diverse families of chemoreceptors in ants should strive to examine a broad range of taxa and use a significantly larger library of odorant stimuli, including but not limited to the hydrocarbons. Future efforts should also strive to incorporate IRs and GRs together with ORs from non-9-exon and species-specific subclades. Given the diverse ecology and extensive chemoreceptor repertoire in ants, addressing this knowledge gap would be a monumental accomplishment. By extending these studies to other species, other chemoreceptor families, and the various subclades within each family, we will develop a better understanding of the evolution of eusociality and the molecular mechanisms involved in social behavior, as well as pave the way for future studies by identifying candidates for gene editing and other targeted molecular approaches.

## 3. Central Olfactory System

### 3.1. An Overview of the Central Olfactory System in Insects

At a cellular level, the fundamental organization of the olfactory system is remarkably similar between vertebrate and insect species [58,59]. Across this broad evolutionary distance, diverse OSNs residing in an aqueous milieu receive chemical messages from the environment, and this information is relayed to the central brain via dedicated axonal tracts, converging on secondary neurons, local interneurons, and glial cells that together constitute the neuropil which forms the stereotypic glomeruli of the vertebrate olfactory bulb ortholog known in insects as the antennal lobe (AL) [60,61]. Until recently, it was doctrine that a single glomerulus was typically innervated by a specific corresponding set of peripheral OSNs, many of which express the same chemoreceptor [62,63]. There may, however, be important exceptions to this rule as emerging studies from *Drosophila* and the yellow fever mosquito *Aedes aegypti* reveal that a single OSN may co-express receptors from different chemoreceptor families and are linked to multiple AL glomeruli [64,65]. In any case, having arrived at their respective (or collective) AL glomeruli, synaptic connections relay information to a collection of secondary glomerular neurons, known in insects as AL projection neurons, which are comparable to vertebrate olfactory bulb mitral and tufted cells. The initial processing of peripheral olfactory information that eventually leads to odorant discrimination and presumably perception occurs through the combinatorial activation of glomeruli that is transformed through integrative (often inhibitory) crosstalk between glomeruli via local interneurons [66,67,68,69,70]. Projection neurons subsequently connect the olfactory bulb or AL to the olfactory cortex and other central brain structures in vertebrates or, in the case of insects, to the mushroom bodies and lateral horn of the protocerebrum [60,61]. In ants and other Hymenoptera, projection neurons are organized into a unique, dual olfactory pathway consisting of a medial and lateral output tract connecting to higher order brain structures which may improve olfactory information processing (Figure 2) [71,72]. These structures are then responsible for more complex cognitive processes. It has been suggested that insect mushroom bodies are responsible for learning and memory [73,74], whereas the lateral horn may play a role in learned and innate behavioral responses [75].

### 3.2. Structure and Function of the Antennal Lobe

While there are indeed many parallels between the insect and vertebrate olfactory systems, there are also notable differences in terms of scale, structure, and function. Mice have well over 1000 olfactory bulb glomeruli which, following the oft-cited, “one-receptor-one neuron-one glomerulus” rule [76,77], derives from a correspondingly similar number of ORs [78,79]. In contrast, *Drosophila* maintain 62 ORs and a comparable number of AL glomeruli [63,80]. As one might expect given their significantly larger OR repertoires, the complexity of ant ALs falls somewhere in between—the clonal raider ant *Ooceraea biroi*, for example, has approximately 500 glomeruli [41] whereas leaf-cutting *Atta vollenweideri* have about 390 among the smaller worker caste and 440 among the larger workers [81]. Importantly, the precise composition of the AL in ants and other eusocial Hymenopteran varies dramatically among different colony members within a given species with respect to age, task, and morphology [81,82,83,84,85]. Previous experience and exposure to different environmental conditions may also lead to changes in glomerular volume, odor coding, and behavior [86]. A distinct group of larger *Camponotus* workers (“majors”) have a correspondingly larger glomerular volume but fewer glomeruli than minor workers [85]. By contrast, larger workers in the leaf-cutting ant *A. vollenweideri* have a greater number of glomeruli than minor workers [81]. Interestingly, high volume macroglomeruli, which are about 9–10 times larger than average glomeruli, have also been identified in the larger worker caste of leaf-cutting ants, and these may be responsible for the detection of trail pheromones [87]. Furthermore, the ALs display profound sexual dimorphisms. In *C. japonicas*, sterile female workers and virgin queens have roughly 430 glomeruli, whereas the AL of males is reduced to only 215 glomeruli [88]. In *C. japonicas*, as well as other Hymenopteran species such as the honeybee *Apis mellifera*, males also have larger macroglomeruli structures, which are thought to be involved in the detection of sex pheromones [45,88,89,90,91]. These male-specific characteristics may reflect their marginalized role as short-lived reproductives. Overall, these changes likely reflect the unique behavioral and reproductive tasks carried out by different members of an ant colony.

The detection of social cues including pheromones and chemical blends such as CHCs distinguish ants and other eusocial species from solitary insects; however, it is noteworthy that odor coding in the AL is conserved across a broad evolutionary distance. Across insects and mammals, for example, the neuronal representation of general odorants among the AL glomeruli are organized and structured around distinguishing features of a given chemical class, including chain length and functional groups [92], and these activation patterns are consistent across members of a given species [72,93]. While glomerular activation patterns for both pheromones and non-pheromone odorants may overlap in ants [72,94], other studies have shown discrete clusters of glomeruli responsible for the detection of alarm and trail pheromone signals [87,95] and this is consistent with studies conducted in moths, which have dedicated macroglomerular complexes involved in the detection of sex pheromones [96] in addition to ordinary glomeruli that process both general plant volatiles and sex pheromones [97].

### 3.3. Olfactory Sensory Neurons and the Ontogeny of the Antennal Lobe

Another notable difference between the insect and vertebrate olfactory systems concerns the relationship between diverse sets of OSNs and the ontogeny of the AL glomeruli. In *Drosophila*, AL development occurs through three phases that begin at the start of pupation when dendrites from second-order projection neurons arrive at stereotypic sites in the brain [98]. In the second phase, OSN axons from peripheral olfactory appendages arrive at target sites in the proto-antennal lobe. This second phase notably occurs prior to OR gene expression and, not surprisingly, there are no significant structural alterations to the glomeruli of *orco* null mutant *Drosophila* [99]. Furthermore, OSNs survive through development but degenerate later in adulthood. This is in contrast to mice and other mammals, where functional ORs and OSNs are required for proper axon targeting [100] and are capable of regeneration [101]. In the final phase, fruit fly projection neurons and the axons from OSNs establish local synaptic connections to the exclusion of neighboring cells to create discrete glomeruli.

Arguably, the most compelling distinction about the olfactory system in ants compared to fruit flies and mosquitoes was the recent observation that Orco is required for the proper development of the AL glomeruli [102,103]. This rather unexpected difference in ant brain development was first described in two parallel reports using CRISPR-Cas9 gene editing to knock out *Orco* in the jumping ant *Harpegnathos saltator* [102] and the clonal raider *O. biroi* [103]. In addition to a profound loss of olfactory sensitivity, as well as the alteration of several behavioral phenotypes that were both anticipated, *orco* mutant ants displayed significant reductions in both OSN populations and the number and volume of AL glomeruli. More recently, AL development in *O. biroi* has been closely examined during the critical two-week pupation period [104]. In contrast with *Drosophila* [99], OR expression occurs much earlier in ant development, before the formation of glomeruli [104]. Indeed, *Orco* expression was high on the first day of pupal development, and almost all of the nearly 500 ant ORs were expressed by day 2 of the pupal stage. Moreover, while Orco is localized to the dendrites and cell bodies of fruit fly OSNs [99], in the clonal raider ant, it is also found in OSN axons and axon terminals in the brain. Here, unilateral antennal ablations (that impact only the ipsilateral half of the bilaterally symmetric ALs) on the first day of pupation resulted in significantly reduced glomeruli in adults. When antennae were ablated later in pupation, development was arrested, but any glomeruli that had already formed survived to adulthood. When antennae were ablated in adult callow workers, AL glomeruli remain for at least two weeks. Taken together, this suggested that *orco* mutants have impaired AL development due to loss of OSNs, which were necessary for the formation of glomeruli but not their maintenance. Curiously, approximately 90 glomeruli survived both the ablation treatment and in the *orco* null mutant [103,104]. The authors suggested these remaining glomeruli may be a more basic template upon which the remainder of the more complex AL forms. 

Developing a topographical map of the AL in ants, as is being done in the honeybee [105], would catalyze the effort to provide a better understanding of odor coding in the glomeruli. These insights may shed light on the role of the mysterious 90 surviving glomeruli, if they have a function at all, and how their development may differ from the remainder of the AL. Ultimately, however, we are left with more questions than answers. This is especially so in light of recent studies in fruit flies and mosquitoes demonstrating that subsets of olfactory neurons co-express ORs, IRs, as well as potentially other receptor classes [64,65]. Such polymodal neurons display non-canonical relationships to the AL that upset the “one receptor-one neuron-one glomerulus rule” and provide a fruitful avenue for future research.

## 4. Genomics, Evolution, and the Regulation of Chemosensory Genes

Over the past decade, considerable progress has been made toward understanding olfactory genomics in eusocial insects [5]. During this time, more than 50 Hymenopteran genomes have come online, and sequencing efforts for many more are currently underway [106,107]. One of the most notable scientific discoveries resulting from this ever-growing repository of genomic data was the identification of significant changes in the chemoreceptor families [21,22,108,109,110,111,112,113,114]. Specifically, there has been a massive expansion of ORs through gene birth-and-death evolution across Apocrita that directly correlates to the degree of eusociality [21,22]. Among these, ants boast the largest number of ORs. Genome sequencing across the evolutionarily basal suborder Symphyta, which is devoid of any eusocial species, has been considerably more limited. One bioinformatics study completed thus far in Symphyta has revealed that the genome of the solitary wheat stem sawfly *Cephus cinctus* has not undergone the same expansion of ORs as seen in Apocrita [115]. A notable exception to the eusocial-driven expansion of OR gene families is the genomes of several species of solitary wasps, including *Nasonia vitripennis* and *Microplitis demolitor*, each of which have more ORs than that of the eusocial honeybee *A. mellifera* [22,116]. Looking at chemoreceptors beyond ORs, the genome of the dampwood termite *Zootermopsis nevadensis* has a greatly expanded family of IRs [117]. Similarly, cockroaches have the largest number of chemoreceptors of any insect species described to date, with massive expansions of the IR and GR families [118]. While insufficient to fully explain the macroevolution of eusociality, the expanded capacity to detect and communicate chemical information likely facilitated the acquisition of the broad range of social behaviors that doubtlessly also provided an adaptive advantage across diverse environments. These early genomic studies in Hymenoptera provided a clear sense of direction for future research spurred by the concurrent development of molecular tools in eusocial insects.

### 4.1. Targeted Gene Editing in Formicidae

As discussed above, the first and arguably most significant Hymenopteran gene-editing accomplishment thus far was carried out in Formicidae using CRISPR-Cas9 technology to knock out *Orco* in two different species of ants, *H. saltator* and *O. biroi* [102,103]. In addition to the neuroanatomical effects, the authors of these two studies reported a number of physiological and behavioral deficits in *orco^−/−^* mutants that would likely impact eusociality. To begin with, in the absence of OR-mediated signaling, social cohesion within colonies was significantly diminished as workers wandered outside of the colony and neglected to engage in brood care. As to be expected, mutant workers displayed a loss of responsiveness to a number of olfactory cues and failed to follow trail pheromones or congregate with nestmates. Mutant workers from both species also had low fecundity. Taken together, these studies were meaningful not only because of the biological insights gleaned but also for their technical merit in extending gene targeting to eusocial insects despite their unique reproductive division of labor.

### 4.2. The Technical Challenges of Gene Editing in Eusocial Hymenoptera

The toolbox available for examining the molecular biology of *Drosophila* as an academic model system has grown immensely since the pioneering genetic studies of Thomas Hunt Morgan at the turn of the 20th century. However, despite the availability of these resources, the transfer of these techniques to ants and other non-model insects has not been as rapid as many investigators had initially expected. For example, in comparison with the now countless numbers of mutant *Drosophila* lines that have been produced, there are (at this writing) only three published studies that have successfully utilized CRISPR-Cas9 or any other type of gene editing in ants [102,103,119]. This is not entirely surprising given the exclusive constraints imposed by the unique reproductive biology and other atypical features of many eusocial Hymenopteran relative to *Drosophila*.

To begin with, it is important to appreciate that, unlike the short and experimentally amenable lifecycle of solitary *Drosophila* and other Diptera (which often can be individually mated), the generation time in ant colonies can be quite long. Ant colonies are typically comprised of one or several extremely long-lived reproductive queens [1]. As the colony matures, diploid virgin queens and haploid males will emerge from the colony to engage in a mating flight before establishing a new colony. Even if an ant colony had a much shorter generation time and was capable of producing many offspring, the reproductive timing of virgin daughters and reproductive males is largely unknown [1]. In addition, ant colonies reared in a lab setting do not always produce reproductives, perhaps in part due to the use of temperature and humidity-controlled incubation chambers (S.T. Ferguson, personal observation). Furthermore, it is currently not possible to identify the subset of embryos that will ultimately develop into either reproductive queens or sterile workers at the very narrowly timed syncytial blastoderm developmental stage required for robust CRISPR-Cas9 injections to target pole cells representing the inherited germlines. While a colony may produce hundreds of millions of eggs over its lifespan [120], which at first might seem ideal for injection-based gene editing, the vast majority of these develop into sterile females. Therefore, as challenging as it may be, although the injection itself may be successful and yield a viable larval-stage transgenic, it is extremely difficult to develop genetic lines, let alone rear sufficient numbers of individuals for studies that involve the collective behavior of a full colony.

Given these challenges, successful gene editing in three ant species must be viewed as exceptional, although it is noteworthy that each of these studies exploited a specific quirk of reproductive biology. For example, after the death of a queen, workers in *H. saltator* colonies compete in a ritualized dueling behavior. The winner of these bouts undergoes a series of physiological changes to become a reproductive gamergate. In the absence of nestmates, segregated workers will also transition to gamergates. Prior to mating, gamergates will lay eggs that develop into males. After mating, they are capable of producing female workers that continue to maintain the colony. Taking advantage of this, investigators designed guide RNAs (gRNAs) targeting the *H. saltator Orco* gene that together with Cas9 protein were microinjected into male embryos. In order to prevent the destruction that typically occurs when manipulated embryos are reintroduced into *H. saltator* colonies, all injected embryos were independently reared outside the colony for 1 month on agar plates. Only after injected embryos had hatched into larvae were they placed into small nests together with a limited number of helper workers that acted as nurses that were required for larval survival. Resultant adults were then outcrossed to produce a mix of mutant and wild-type male offspring [102]. The genotype of these males was identified nonlethally by sequencing tissue samples obtained from the wing. Through a series of successive crosses extending over more than one year, mutant males were used to eventually establish homozygous mutant lines. The clonal raider ant, *O. biroi*, has a fundamentally different reproductive system, characterized by queenless colonies, in which workers reproduce parthenogenetically. Here, mutant lines (also targeting *O. biroi Orco*) were established from injected individual embryos without the need for extensive crosses [103]. More recently, CRISPR-Cas9 has been used to induce somatic mutations in the fire ant, *Solenopsis invicta* [119]. In this case, rather than attempt to generate stable mutant lines, the authors directly injected worker embryos with Cas9 together with gRNAs targeting *GP-9*, which encodes an odorant binding protein suspected of being associated with colony form, and *Sinv-spitz*, which was thought to be involved in establishing larval oenocytes. While these investigators used PCR to successfully establish some molecular evidence of gene targeting, they were unable to observe any physiological or behavioral phenotypes [119]. At the end of the day, the absence of phenotypic effects raises questions as to the utility of an individual level approach for examining the biology of eusocial ants that function within complex colonies that have often been described as “superorganisms” [121].

### 4.3. Innovative Variations and Alternatives to Gene Editing

These gene-editing studies highlight the ingenuity of the investigators, the creativity often required of scientific endeavors, and, in the case of *H. saltator* and *O. biroi*, have provided unique insights into the olfactory system and social behavior of ants. Without diminishing these accomplishments, one might raise a caveat in that the methods employed rely on the decidedly atypical reproductive biology of *H. saltator* and *O. biroi*, both of which are not representative of most ant species. Therefore, it is reasonable to suggest that the existential challenges associated with gene editing in eusocial species remain to be addressed in a more direct, generalizable way. One potential solution may be through an innovative approach to insect gene editing that has been termed “ReMOT Control”, short for Receptor-Mediated Ovary Transduction of Cargo [122]. Here, CRISPR-Cas9/gRNA machinery is delivered to developing eggs during vitellogenesis using modified yolk protein precursors that are transported from the hemolymph into the ovaries. Indeed, this method has proven successful across a broad range of insect species [123,124,125]. Another approach might be to deliver the CRISPR-Cas9/gRNA complex using transfected sperm, a protocol for which has been successfully developed in birds using a cationic lipid-based chemical transfectant [126], by artificially inseminating virgin queens [127].

While there is no argument that the generation of *orco^−/−^* mutants in *H. saltator* and *O. biroi* represented a quantum leap in olfaction studies in ants, an additional and often-overlooked consideration that is salient for gene-editing studies in any system is the potential for off-target effects. Indeed, the catastrophic changes to the AL during development represent a nontrivial confounding variable. Taken together with more recent efforts [104], these studies suggest that OR function plays a necessary role in a variety of social behaviors that contribute to the evolutionary success of these insects. That said, it is not possible to distinguish whether the behavioral phenotypes observed in these mutants are the result of the loss of olfactory signaling from the antennae, the large defects in the AL, or any number of potential changes encountered during an altered developmental program.

To address this confounding factor, we recently took advantage of a set of recently identified, novel pharmacological agents that acutely and selectively modulate Orco activity to examine the role of OR signaling in nestmate recognition [128]. These compounds include an allosteric agonist, an allosteric antagonist, and a physiologically and pharmacologically inert analog control, all of which can be applied as volatiles to wild-type adult ants. This method provided a potentially superior alternative to genetic engineering in that it disrupts olfactory signaling at a discrete time point in wild-type adults that had a normal developmental trajectory and were not subject to nearly impossible-to-rule-out off-target pleiotropic effects. Administration of an Orco antagonist conclusively demonstrated that OR-signaling is necessary for eliciting aggression toward non-nestmates, and moreover that the lack of familiar nestmate signals is not sufficient to elicit aggression. Parallel studies with the Orco agonist indicated that a mismatch between an olfactory cue and an endogenous template for nestmate odor profiles is also not sufficient to elicit aggression. Instead, aggression toward non-nestmates requires the OR-dependent detection of a precise chemical trigger present on the cuticle of a non-nestmate foe. Importantly, because Orco is highly conserved across insect species, this method can readily be applied to diverse ant taxa. However, the broad utility of this approach is limited by lack of similar pharmacological agents against other cellular and molecular targets.

### 4.4. Advances in Epigenetic Engineering

Beyond gene editing, there have been other major technical advances in genomic myrmecology. Most notably, innovations in examining the epigenetics of ants. Eukaryotic DNA is compacted into chromatin complexes enveloping histone protein nucleosome octamers which are then altered through histone post-translational modifications (hPTMs) that directly influence the regulation of gene expression by altering the structure and accessibility of the DNA-protein complex [129,130]. Methylation of cytosine nucleobases within DNA may also regulate gene expression and result in static ‘imprinting’ that impacts discrete genes as well as entire chromosomes [131]. Importantly, these processes, along with other epigenetic modifications, are likely to play a central role in the regulation of olfactory gene expression. Lysine methylation of histone 3 in the fruit fly, for example, determines OR gene expression by silencing the expression of all but one receptor [132,133,134]. The development of novel approaches to modify this “histone code” may, therefore, represent an important avenue for studies of olfaction in ants and other insects.

### 4.5. Artificially Induced Histone Modifications Dramatically Alter Ant Behavior

While the genome of an individual organism is static, the methylome may vary across cells, tissues, and organisms. Genome-wide studies have now broadly characterized DNA methylation patterns and histone modifications in *C. floridanus* reproductive and morphological castes [135,136]. DNA methylation mapping revealed surprisingly few distinctions between majors and minors [114]. Furthermore, in these studies, gene expression did not seem to strongly correlate with DNA methylation. However, major and minor workers exhibit caste-specific enrichment of hPTMs. Acetylation of lysine 27 of histone H3 proteins (H3K27ac), which is typically associated with transcriptional activation, correlates with caste-specific gene expression patterns [135]. In particular, binding sites for histone acetyltransferase (HAT) and CREB binding protein (CBP)—both involved in histone acetylation and transcriptional activation—displayed the greatest variation among castes. Therefore, hPTMs may play a critical role in establishing transcriptional differences between morphological castes and task groups. For example, previous studies have demonstrated that, in *C. floridanus*, minor workers carry out the majority of the foraging, while majors forage very little [137]. However, microinjection of a histone deacetylase inhibitor (HDACi)—a class of epigenetic modifying drug that fosters chromatin acetylation—caused majors to engage in robust, minor-like foraging activity. This effect was inhibited when young majors were co-injected with both an HDACi and a HAT inhibitor that would be expected to have opposing effects [137]. These results support the hypothesis that histone acetylation regulates the behavioral differences between foraging minors and non-foraging majors.

The use of small-molecule histone-modifying pharmacological compounds offers yet another powerful molecular tool for the epigenetic study of myrmecology. While these studies do not directly address the role of olfactory signaling in directing worker behavior, the observation that histone modifications play an important role in the modulation of insect OR gene expression [132,133,134] makes a strong case for its involvement. Therefore, future studies might explore the connection between the epigenetic regulation of chemoreceptor gene expression and social behavior in ant colonies using these and other molecular techniques.

## 5. Conclusions

Beginning with the availability of the first two ant genomes [20], there has been considerable progress in the field of olfactory myrmecology. These studies have illuminated a high degree of complexity in the olfactory system of ants which differentiates these insects from other, more traditional model systems such as the fruit fly (Figure 2). Moreover, in contrast to solitary organisms, eusocial ants engage in olfactory-driven collective behaviors within colonial societies (Figure 1) that may drive the evolution of fundamental differences in how the olfactory system develops and ultimately feeds into the unique behavioral repertoire of ants and eusocial insects. While mindful of the caveat that advances in molecular techniques in eusocial insects have thus far been limited to laboratory studies that constrain inferences regarding the salient natural ecology, these studies provide a foundation for future studies to explore the relationship between the complex ant olfactory system and the social environment of the colony. Despite the considerable progress that has been made over the past decade, marked by a series of high-impact studies that have received attention in the scientific community and beyond, much work remains to be done. These efforts, while challenging, are exceptionally meaningful, and eusocial insects are quickly becoming tractable model systems with a growing repository of tools. Taken together, these studies support the continuation of basic research as a means to uncover more fundamental principles in diverse biological systems that transcend seemingly disparate taxa that may have important implications for our understanding of the biology and even the sociology of many vertebrate species, including most notably our own, *Homo sapiens*.

## Figures and Tables

**Figure 1 insects-12-00252-f001:**
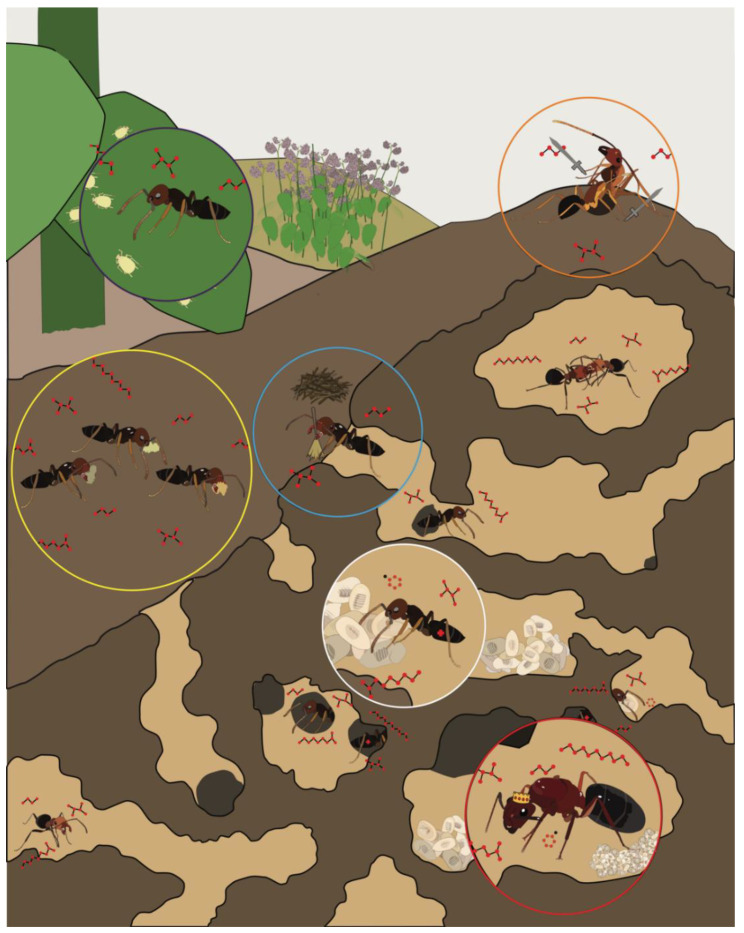
The regulation of complex social behaviors in ant colonies relies on the transmission and detection of chemical information among discrete castes and worker groups, including reproductive “queen(s)” (red circle), brood-care “nurses” (white circle), cleaning “midden workers” (blue circle), food collecting “foragers” (yellow circle), aphid-tending “farmers” (purple circle), and colony defending “soldiers” (orange circle).

**Figure 2 insects-12-00252-f002:**
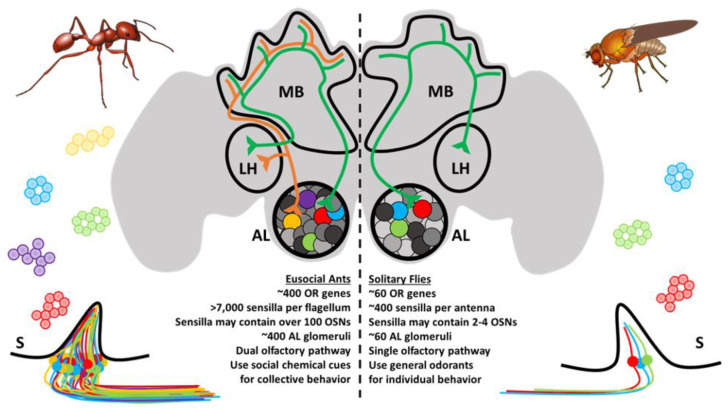
Ants and other eusocial Hymenopteran have a remarkably complex olfactory system relative to fruit flies and other solitary insects. Ant sensilla (S) may contain OSNs that are up to two orders of magnitude more numerous than those of the fruit fly. The antennal lobe (AL) glomeruli are also more numerous and follow a different developmental trajectory. Odor processing in higher-order brain structures, including the mushroom bodies (MB) and lateral horn (LH), occurs through a novel dual-olfactory processing pathway.

## Data Availability

Not available.

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
