# Peer review of "Advances in the Study of Olfaction in Eusocial Ants"

_insects, 2021, doi:10.3390/insects12030252_

Round 1

Reviewer 1 Report

insects-1130872
Advances…
Ferguson et al.

General

An excellent review. The outlook into genomes, gene editing/genetic engineering is an original and engaging way of rounding up with "projections into the future".

The excursion into vertebrate is admittedly concise, but fails to deliver important insights and does not really add to this review.

Otherwise, I have but a few minor comments.

line 31. taxon

33. dominate / significant proportion - rephrase

218. The following para does not really deliver insights into these "parallels"

287. check for use of orco/Orco, depending on context, throughout.

313. insofar?

350. OR-coreceptor Orco (?) (introduce Orco/abbreviation earlier)

Author Response

line 31. taxon

Authors Response: The revised text now reads: “… ants are an extraordinarily successful insect taxon…”

  1. dominate / significant proportion – rephrase

Authors Response: We have removed this text. Now reads: “From an evolutionary perspective, ants are an extraordinarily successful insect taxon that are globally pervasive and comprise more than a quadrillion individuals.”

  1. The following para does not really deliver insights into these "parallels"

Authors Response: We have changed the section subheading to de-emphasize the comparison to vertebrates, which serves as a useful analogy but is not, as the reviewer notes, the primary focus. Now reads: “3.1. An Overview of the Central Olfactory System in Insects”

  1. check for use of orco/Orco, depending on context, throughout.

Authors Response: We have clarified and corrected the use of orco/Orco throughout the text to conform to established conventions in the fly/mosquito research communities (https://wiki.flybase.org/wiki/FlyBase:Nomenclature). The protein is denoted by the use of “Orco”, the wild-type gene is now denoted by the use of “Orco”, and the null mutant is now denoted by the use of “Orco-/-”.

  1. insofar?

Authors Response: The revised text now reads: “…would catalyze the effort to provide a better understanding of odor coding in the glomeruli.”

  1. OR-coreceptor Orco (?) (introduce Orco/abbreviation earlier)

Authors Response: The revised text now reads: “…using CRISPR-Cas9 technology to knock out Orco in two different species of ants…”  Please note that the Orco abbreviation was already introduced in the text previously, in the first paragraph of section 2. The Peripheral Olfactory System.

Reviewer 2 Report

In this manuscript, the authors review advances in the study of olfaction in eusocial ants by focusing on molecular olfaction and by highlighting potential avenues for future research. The paper is well written and will be an important contribution to the olfactory research community, to everybody interested in social behavior, and to biologists at large.

I have only a few minor concerns that might be addressed to improve the manuscript before its publication in Insects.

  • The manuscript aims at describing advances in the study of olfaction in eusocial ants in general. However, the authors focused mainly on the processing of CHC. The manuscript would gain by clarifying that olfaction for an ant is more than the perception of cuticular profiles and social pheromones. Especially at the central level, the work should describe studies on general odorant coding in ants and the similarity/differences found in comparison to other insect species (i.e. Dupuy et al. 2010; Kuebler et al. 2010; Zube et al. 2008; Galizia et al. 1999; etc…).

  • The figures are lacking legends. The authors should add a description of the two figures, especially of the colored circles of Figure 1 and of the neuron types, the abbreviations and the olfactory information processing in Figure 2. Another comment on Figure 2: Why does this Figure appear before paragraph 2.1 and not when the Figure 2 is mentioned, in paragraph 3? It should be moved near the citation.

  • The references are not well formatted. The authors should pay attention to Insects‘ reference format (year in bold…) and to the solitary lines.

Other minor points

  • L.21: Please remove a space before “Despite…”
  • L.109-110: “… tastants as well as carbon dioxide.” The authors should mention that this finding has only been made in drosophila.
  • L.113-114: Same comment as L.109. The authors should mention that IRs have been found to be sensitive to acids and aldehydes only in drosophila for now.
  • L.202-204: Please mention to which odorants the receptors belonging to the H, V and E subfamilies are sensitive to.
  • L.225: Please remove a space before “…vertebrate…”
  • L.253: The authors should mention the number of ORs found in drosophila.
  • L.255-256: Only one example is given for the number of glomeruli found in eusocial ants. Can you please give more examples?
  • L.262-263: “larger workers in the leaf-cutting ant Atta vollenweideri have a greater number of glomeruli”. Than minor workers? Please add details.
  • L.263: The authors mention the term macroglomeruli but do not define it. Please add a short definition of this term.
  • L.269: “be involved in the detection of sex pheromones.” Please add in Hymenopteran species.
  • L.302: “Here, unilateral antennal ablations (that impact only the contralateral half of the bilaterally symmetric ALs) on…”. Do the authors mean ipsilateral half?
  • L.468-469: The authors should also mention that the strong limit of the pharmacological approach is that it cannot be specific of a particular OR (or chemosensory receptor in general) and thus strongly limits the questions asked.
  • L.484-520: This paragraph is too long and should be shortened as the link with olfaction was never demonstrated (as mentioned by the authors L.523).

Author Response

The manuscript aims at describing advances in the study of olfaction in eusocial ants in general. However, the authors focused mainly on the processing of CHC. The manuscript would gain by clarifying that olfaction for an ant is more than the perception of cuticular profiles and social pheromones.

Authors Response: The first line of the revised second paragraph now has additional text that reads: “Ants communicate with one another by exchanging an array of chemical messages that include general odorants common to many insects as well as the social pheromones and other chemical blends that distinguish ants from other, solitary insects.”

We have also made the following addition to the revised text of the third sentence of the second paragraph to emphasize that CHCs are being used as just one example. The revised text now reads: “Complex blends of cuticular hydrocarbons (CHCs), as an example…”

In addition, please note that the third paragraph provides examples of additional odorants above and beyond CHCs.

Especially at the central level, the work should describe studies on general odorant coding in ants and the similarity/differences found in comparison to other insect species (i.e. Dupuy et al. 2010; Kuebler et al. 2010; Zube et al. 2008; Galizia et al. 1999; etc…).

Authors Response: We have added a paragraph to section 3.2 of the revised text that reads: “The detection of social cues including pheromones and chemical blends such as CHCs distinguish ants and other eusocial species from solitary insects; however, it is noteworthy that odor coding in the AL is conserved across a broad evolutionary distance. Across insects and mammals, for example, the neuronal representation of general odorants among the AL glomeruli are organized and structured around distinguishing features of a given chemical class, including chain length and functional groups [82], and these activation patterns are consistent across members of a given species [72,93]. While glomerular activation patterns for both pheromones and non-pheromone odorants may overlap in ants [94], other studies have shown discrete clusters of glomeruli responsible for the detection of alarm and trail pheromone signals [88,95] and this is consistent with studies conducted in moths, which have dedicated macroglomerular complexes involved in the detection of sex pheromones [96] in addition to ordinary glomeruli that process both general plant volatiles and sex pheromones [97].”

The figures are lacking legends.

Authors Response: The legend to Figure 1 reads: “The regulation of complex social behaviors in ant colonies relies on the transmission and detection of chemical information among discrete castes and worker groups, including reproductive “queen(s)” (red circle), brood-care “nurses” (white circle), cleaning “midden workers” (blue circle), food collecting “foragers” (yellow circle), aphid-tending “farmers” (purple circle), and colony defending “soldiers” (orange circle).”

As orginally submitted the legend to Figure 2 reads: “Ants and other eusocial Hymenopteran have a remarkably complex olfactory system relative to fruit flies and other solitary insects. Ant sensilla (S) may contain OSNs that are up to two orders of magnitude more numerous than those of the fruit fly. The antennal lobe (AL) glomeruli are also more numerous and follow a different developmental trajectory. Odor processing in higher-order brain structures including the mushroom bodies (MB) and lateral horn (LH) occurs through a novel dual-olfactory processing pathway.”

Another comment on Figure 2: Why does this Figure appear before paragraph 2.1 and not when the Figure 2 is mentioned, in paragraph 3? It should be moved near the citation.

Authors Response: Inasmuch as Figure 2 is first cited in the second paragraph of section 2, we therefore did not reposition this figure within the text. In any case we are happy for the editor to position the figure to where-ever they please.

The references are not well formatted. The authors should pay attention to Insects‘ reference format (year in bold…) and to the solitary lines.Authors Response: We used the MDPI EndNote template (downloaded online) and respectfully believe the references to be formatted properly (year in bold for journal articles but not for other sources such as books or book chapters). That said, we have corrected several inadvertent errors, including the solitary lines which were the result of an extra space in the DOI of the reference (28, 32, 38, 113), we corrected counts of total page numbers (1, 2), removed an erroneous DOI that had merged with an ISBN number (17), reformatted the article title and journal name which were previously in full caps (120). We also removed an improper citation from Vowles, D (previously 73).

L.21: Please remove a space before “Despite…”

Authors Response: We have removed the space before “Despite…”

L.109-110: “… tastants as well as carbon dioxide.” The authors should mention that this finding has only been made in drosophila.

Authors Response: The revised text now reads: “…and, at least in Drosophila, are responsible for…”

We have also deleted an additional space before the word “Beyond” in the following sentence.

L.113-114: Same comment as L.109. The authors should mention that IRs have been found to be sensitive to acids and aldehydes only in drosophila for now.

Authors Response: The revised text now reads: “…are, in Drosophila, responsible for the detection of…”

L.202-204: Please mention to which odorants the receptors belonging to the H, V and E subfamilies are sensitive to.

Authors Response: We have rephrased this sentence for clarity. Now reads: “…all of which respond to long-chain alkanes.”

L.225: Please remove a space before “…vertebrate…”

Authors Response: We have removed the space and also added a comma after “Until recently, …” in the following sentence.

L.253: The authors should mention the number of ORs found in drosophila.

Authors Response: The revised text now reads: “In contrast, Drosophila maintain 62 ORs and a comparable number of AL glomeruli.”

L.255-256: Only one example is given for the number of glomeruli found in eusocial ants. Can you please give more examples?

Authors Response: The revised text now reads: “the clonal raider ant Ooceraea biroi, for example, has approximately 500 glomeruli [41] whereas leaf-cutting Atta vollenweideri have about 390 among the smaller worker caste and 440 among the larger workers [82].”

Note that an additional example was also already provided in more detail at the end of this same paragraph.

L.262-263: “larger workers in the leaf-cutting ant Atta vollenweideri have a greater number of glomeruli”. Than minor workers? Please add details.

Authors Response: The revised text now reads: “By contrast, larger workers in the leaf-cutting ant Atta vollenweideri have a greater number of glomeruli than minor workers.”

L.263: The authors mention the term macroglomeruli but do not define it. Please add a short definition of this term.

Authors Response: We have added additional detail. Now reads: “Interestingly, high volume macroglomeruli, which are about 9-10 times larger than average glomeruli…”

L.269: “be involved in the detection of sex pheromones.” Please add in Hymenopteran species.

Authors Response: The revised text now reads: “In Camponotus japonicas, as well as other Hymenopteran species such as the honeybee Apis mellifera, males also have larger macroglomeruli structures…”

L.302: “Here, unilateral antennal ablations (that impact only the contralateral half of the bilaterally symmetric ALs) on…”. Do the authors mean ipsilateral half?

Authors Response: We have changed “contralateral” to “ipsilateral”.

L.468-469: The authors should also mention that the strong limit of the pharmacological approach is that it cannot be specific of a particular OR (or chemosensory receptor in general) and thus strongly limits the questions asked.

Authors Response: We fully agree with the reviewer regarding this caveat and respectfully believe this limitation was addressed in the concluding sentence of section 4.3, that reads: “However, the broad utility of this approach is limited by lack of similar pharmacological agents against other cellular and molecular targets.”

L.484-520: This paragraph is too long and should be shortened as the link with olfaction was never demonstrated (as mentioned by the authors L.523).

Authors Response: We have merged the first two paragraphs of the revised section 4.5 together and deleted an excess of detail regarding the epigenetic methods to streamline this section.
